# Color, Antioxidant Capacity and Flavonoid Composition in *Hibiscus rosa*-*sinensis* Cultivars

**DOI:** 10.3390/molecules28041779

**Published:** 2023-02-13

**Authors:** Jesica J. Mejía, Lady J. Sierra, José G. Ceballos, Jairo R. Martínez, Elena E. Stashenko

**Affiliations:** Center for Chromatography and Mass Spectrometry (CROM-MASS), Research Center for Biomolecules (CIBMOL), Universidad Industrial de Santander, Bucaramanga 680002, Colombia

**Keywords:** *Hibiscus rosa-sinensis*, color, flavonoids, antioxidant capacity

## Abstract

*Hibiscus rosa-sinensis* plants are mainly cultivated as ornamental plants, but they also have food and medicinal uses. In this work, 16 *H. rosa-sinensis* cultivars were studied to measure their colorimetric parameters and the chemical composition of hydroethanolic extracts obtained from their petals. These extracts were characterized using UHPLC-ESI^+^-Obitrap-MS, and their antioxidant activity was evaluated using the ORAC assay. The identified flavonoids included anthocyanins derived from cyanidin, glycosylated flavonols derived from quercetin and kaempferol, and flavan-3-ols such as catechin and epicatechin. Cyanidin-sophoroside was the anthocyanin present in extracts of lilac, pink, orange, and red flowers, but was not detected in extracts of white or yellow flowers. The total flavonol concentration in the flower extracts was inversely proportional to the total anthocyanin content. The flavonol concentration varied according to the cultivar in the following order: red < pink < orange < yellow ≈ white, with the extract from the red flower presenting the lowest flavonol concentration and the highest anthocyanin concentration. The antioxidant activity increased in proportion to the anthocyanin concentration, from 1580 µmol Trolox^®^/g sample (white cultivar) to 3840 µmol Trolox^®^/g sample (red cultivar).

## 1. Introduction

*Hibiscus* is a genus that belongs to the Malvaceae family and comprises more than 200 species, among which are herbaceous, annual or perennial plants. Plants of this genus are native to Southeast Asia; they grow in warm environments and tropical and subtropical regions, produce large and magnificent flowers of various colors, and some are used for ornamental purposes [1]. The *Hibiscus rosa-sinensis* flower is Malaysia’s national flower [2]. In South Asia, these flowers are used in traditional medicine to relieve bronchial catarrh and diarrhea [3]. In Colombia, these plants are known as “cayenos”, and they are cultivated for ornamental purposes due to their brightly colored flowers. The mechanism that regulates the biosynthesis of pigments responsible for *H. rosa-sinensis* petal colors is not known in detail [4]. In nature, colors are important signals that show warnings (predator deterrence) or rewards (pollinator attraction). Betalains, carotenoids, flavonoids, and anthocyanins confer color to flowers and fruits [5,6]. The yellow and red colors are associated with the biosynthetic pathways of betalains and carotenoids, whereas the anthocyanin biosynthesis pathway generates a wide variety of colors ranging from yellow (λ = 480 nm) to orange, blue and red (λ = 730 nm) [6]. Several flavonoid derivatives participate in this biosynthetic pathway, from colorless compounds such as flavonols to colored pigments such as anthocyanins, tannins and proanthocyanidins [7].

In addition to giving color to flowers and fruits, flavonoids have different biological properties, including antioxidant [8], anticancer [9], antiswelling [10], etc. Phytochemical studies on different parts of the *H. rosa-sinensis* plant [11,12,13,14,15,16,17] highlight the presence of alkaloids, coumarins, tannins, saponins and flavonoids. Some studies [18,19,20,21] have described the presence of various flavonoids, such as catechin, epicatechin, cyanidine-3-glucoside, quercetin, kaempferol and their glycosylated derivatives. Shen et al. [19], Nakamura et al. [22] and Rengarajan et al. [23] isolated quercetin-sophoroside, cyanidin-sophoroside, and hibiscetin-3-glucoside, respectively, from *H. rosa-sinensis* flower extracts. Zaki et al. [24] isolated luteolin-7-glucoside from the hydromethanolic extract of *H. rosa-sinensis* leaves.

Until now, no studies have been found that relate flavonoid content with color and antioxidant capacity in *H. rosa-sinensis* cultivars (Figure 1). The objective of this study was to determine a potential relationship between the type and quantity of flavonoids and the color and antioxidant capacity of extracts from *H. rosa-sinensis* flowers. In total, 20 flavonoids present in the petal extracts of 16 *H. rosa-sinensis* cultivars of different colors were identified and quantified by UHPLC-ESI+-Orbitrap-MS. The antioxidant capacity of all extracts was measured using the ORAC method, which permitted the selection of specific cultivars as potential sources of natural ingredients.

## 2. Results

### 2.1. Identification by UHPLC-ESI+-Orbitrap-MS of Flavonoids Present in the Extracts of H. rosa-sinensis Flowers

UHPLC-ESI^+^-Orbitrap-MS analyses allowed the detection of 20 flavonoids in the flower extracts of 16 *H. rosa-sinensis* cultivars (Table 1). Three anthocyanins, eleven flavonols, two flavan-3-ols, two flavanonols, one flavone, and one flavanone were identified. Catechin and epicatechin were the only aglycones, and the remaining detected compounds were glycosylated derivatives of the corresponding flavonoids. Figure 2 shows some chromatographic profiles obtained by LC/MS (Orbitrap) of the hydroalcoholic extracts of four *H. rosa-sinensis* cultivars. Quercetin-hexuronide-hexoside, cathechin, quercetin-dihexoside, kaempferol-deoxyhexoside-hexoside, kaempferol-deoxyhexoside, quercetin-hexuronide, quercetin-3-glucoside, and apigenin-hexuronide were the compounds common to all *H. rosa-sinensis* hydroalcoholic extracts examined. The identification of the chromatographic peaks appearing in Figure 1 is presented in Table 1.

The analysis of the mass spectra obtained with different energies (10, 20, 30 or 40 eV) in the HCD (Higher Energy Collision Dissociation Cell) was based on the study of the characteristic fragmentation patterns for flavonoids (Figure 3), which include the typical loss of molecules such as hexoses (C_6_H_10_O_5_), hexuronides (C_6_H_8_O_6_), H_2_O and CO, in addition to the rupture suffered by the C ring.

Three characteristic fragments were observed in the mass spectra (HCD, 20 eV) of the anthocyanins (Table 1), of which the most abundant was the cyanidin aglycone (*m/z* 287.05515, 100%), produced by the loss of the respective sugars of the protonated molecule. The fragments at *m/z* 137.02336 (3–66%) and at *m/z* 121.02866 (1%) were generated by breaking the C(9)–O(1) and C(2)–C(3) bonds, C(9)–O(1) and C(3)–C(4) of the pyrilium cation, respectively. Barnes and Schug [25] reported the same fragments in the mass spectrum of cyanidin-3,5-glucoside.

Naringenin-hexuronide was the only flavanone detected in the *H. rosa-sinensis* extracts. In its mass spectrum (HCD, 10 eV), the loss of hexuronide (C_6_H_8_O_6_) was observed, accompanied by the formation of the product ion [(M+H)-C_6_H_8_O_6_]^+^ at *m/z* 273.07559 (100%), in addition to the loss of ring B [(M+H)-C_12_H_14_O_7_]^+^, leading to the formation of the ion at *m/z* 179.03374, with low intensity (1%). Cleavage in the C ring generated the product ion [(M+H)-C_6_H_8_O_6_-C_8_H_8_O]^+^ at *m/z* 153.01829 (8%).

Flavonols, glycosylated derivatives of quercetin and kaempferol, were the most common flavonoids detected in *H. rosa-sinensis* extracts (Table 1). The loss of a hexose molecule (C_6_H_10_O_5_) was observed in the mass spectra (HCD, 10 or 20 eV) of the flavonols kaempferol-3-glucoside, quercetin-3-glucoside, kaempferol-deoxyhexoside-hexoside, kaempferol-dihexoside, kaempferol-hexuronide-hexoside, quercetin-hexuronide-hexoside, and quercetin-trihexoside, accompanied by the formation of product ions [(M+H)-C_6_H_10_O_5_]^+^ at *m/z* 287.05566 (100%), *m/z* 303.05005 (100%), *m/z* 433.11310 (20%), *m/z* 449.10754 (20%), *m/z* 463.08743 (100%), *m/z* 479.08209 (100%) and *m/z* 627.15594 (3%), respectively. The protonated quercetin-hexuronide and kaempferol-hexuronide molecules underwent ruptures and losses of a hexuronide molecule [(M+H)-C_6_H_8_O_6_], accompanied by the appearance of the product ions at *m/z* 287.05490 (45%) and at *m/z* 303.04968 (100%), respectively.

Diglycosylated and triglycosylated flavonols were also detected in the extracts, with possible R_2_ and R_5_ substituent groups that, when fragmented, lose neutral sugar molecules, generating the respective aglycones. The flavonols presented a rearrangement in the C ring, generating the fragment at *m/z* 153.01836 with relatively low intensity (1–15%) [33,34].

Apigenin-hexuronide (Figure 2, peak N° 20) was the only flavone detected in the *H. rosa-sinensis* extracts. In its mass spectrum (HCD, 20 eV), the loss of hexuronide (C_6_H_8_O_6_) was observed, accompanied by the formation of the product ion [(M+H)-C_6_H_8_O_6_]^+^ at *m/z* 271.05997 (45%). As products of ring C cleavage, the fragments [(M+H)-C_6_H_8_O_6_-C_8_H_6_O]^+^ were generated at *m/z* 153.01824 (C_7_H_5_O_4_^+^, 17%) and [(M+H)-C_6_H_8_O_6_-C_8_H_6_O_3_]^+^ at *m/z* 121.02857 (1%), previously reported by several authors [33,34,35].

### 2.2. Relationship of Chemical Composition with Color and Antioxidant Capacity in the H. rosa-sinensis Cultivars under Study

Table 2 presents the total flavonoid contents (mg/g) according to each subfamily, the colorimetric parameters (L*, a* and b*) of the CIELAB system, and the chromatic representations of the RGB system for the 16 flower cultivars, together with their extract ORAC antioxidant capacity. The antioxidant activity of *H. rosa-sinensis* flower extracts has been determined using several in vitro methods [36]. In this research, the ORAC method was used because in the hydrogen-atom transfer mechanism the peroxyl radicals are used, which predominate in food and biological systems [37]. The codes of the different flowers are shown in Figure 1. Table 3 reports the content of phenolic compounds (mg/g) in *H. rosa-sinensis* flower hydroalcoholic extracts of the different cultivars studied in this work.

## 3. Discussion

Cyanidin-sophoroside was the major component in the red flower extracts (Table 3). Flavonols (quercetin-hexuronide-hexoside, quercetin-dihexoside, quercetin-hexuronide, quercetin-3-glucoside, kaempferol-desoxyhexoside-hexoside, and kaempferol-desoxyhexoside) were found in the *H. rosa-sinensis* extracts studied, but in different amounts (mg/g), depending on the color of the cultivar. Quercetin-3-glucoside was the major component in flower extracts from pink (Table 3), orange (Table 3) and yellow (Table 3) cultivars. In their studies on *H. rosa-sinensis* cultivars, Faten et al. [11], Garg et al. [12], Patel et al. [13], Rao et al. [14], and Wahid et al. [15] reported alkaloids, coumarins, tannins, saponins and flavonoids; dos Santos et al. [18], Shen et al. [19], Li et al. [20], and Pascoal et al. [21] reported the presence of catechin, epicatechin, quercetin, kaempferol and some glycosylated derivatives.

Shen et al. [19] isolated quercetin-sophoroside from the hydroethanolic extract of *H. rosa-sinensis* flowers and determined its therapeutic potential against Alzheimer’s disease. The authors demonstrated that this glycosylated flavonol decreased memory impairment and improved learning in mice with scopolamine-induced cognitive dysfunction, and then assumed that quercetin-sophoroside could be used as a neuroprotectant. In the same investigation [19], due to the lack of a certified reference substance, quercetin-dihexoside was only tentatively identified.

This study detected cyanidin-3-glucoside, described by Li et al. [20], cyanidin-sophoroside (found at high concentrations, 21 mg/g extract, in the red flower extract, Table 3), which had been isolated for the first time by Nakamura et al. [22] from the *H. rosa-sinensis* flower extract and characterized by NMR, and cyanidin-sambubioside, which had been reported in *H*. *sabdariffa* extracts [28]. The phenolic content of *H. rosa-sinensis* red flower extracts was 60% cyanidin-sophoroside. Although there is a large variability in the number of flowers per plant, an average of 400 harvested flowers per year per shrub can be used to estimate a potential production of this anthocyanin-rich extract. A density of 1600 plants/ha and an average 10 g flower weight would lead to an annual production of 6.4 ton of flowers per ha. The solvent extraction yield was calculated by dividing the hydroethanolic extract weight by the plant material weight. The yield obtained in this study was 23 ± 2%(*w*/*w*). Thus, a hectare of red *H. rosa-sinensis* plants would permit the production of 1.4 ton per year, of an extract containing anthocyanins, flavan-3-ols and flavanonols, which may be useful as ingredients for pharmaceutical and cosmetic products. Anthocyanins, flavonols, flavan-3-ols, flavononols and flavones contribute to the antioxidant activity of the extracts of many plant species (Table 4). Zaki et al. [24] isolated luteolin-7-glucoside from the hydromethanolic extract of *H. rosa-sinensis* leaves and evaluated its antioxidant and hypoglycemic effects; the authors reported that in diabetic rats treated with the hydromethanolic extract of *H. rosa-sinensis*, the hypoglycemic and hypocholesterolemic effects decreased, which led to hepatic improvement. Among the components found in the cultivars under study, luteolin-7-glucoside was not detected, but its isomer, kaempferol-3-glucoside, was confirmed.

Figure 4 shows the 3D scatter bubble plots with color scale, where the results of the CIELAB colorimetric coordinates are shown. The size of the bubble represents the antioxidant activity: the larger the bubble size, the higher the ORAC value. The antioxidant activity of the floral extracts varied from 1580 to 3840 μmol Trolox^®^/g of extract; the bubble of the white cultivar (Figure 1P, 1580 ± 41 μmol Trolox^®^/g of extract) was the smallest, and the bubble of the red cultivar (Figure 1A, 3840 ± 74 μmol Trolox^®^/g of extract) was the largest, with the highest antioxidant activity value. The color scale changed, depending on the extract of the cultivar analyzed, for a total of six colors, namely, white, lilac, yellow, orange, pink and red.

Figure 4A,B show that the bubble corresponding to the extract of the dark red flower is located at a greater distance from the other red bubbles because it presents lower L* values (28) compared to the other red bubbles from the red group (L* values 54–61). The lighter the color of the flower (white, lilac and yellow cultivars), the larger the L* values were, compared to the rest of the flowers studied. These results agree with those reported by Wan et al. [44], who showed that at higher concentrations of pelargonidin-3,5-diglucoside in rose petals, the L* value was lower, thus concluding that the red color of rose petals was due to a high anthocyanin content.

Anthocyanins confer colors to flowers and fruits that can vary from orange to red, pink and purple to blue [45]. Their color is related to the structure, e.g., the number of hydroxyl groups, their degree of methylation, the nature or number of sugar molecules attached to the aglycone, the pH, and the copigmentation that can result from various intermolecular associations, between two identical anthocyanins (self-association) or between an anthocyanin and other phenolic compounds (intermolecular copigmentation) [46], and the formation of π–π-type complexes with a metal ion. Gordillo et al. [47] suggested that flavonols and flavan-3-ols, despite being colorless compounds, may play an important role in color generation through copigmentation with anthocyanins, which would increase the stability of the latter. Bakowska et al. [48] reported the formation of complexes of cyandin-3-glucoside and flavonoids isolated from *Scutellaria baicalensis*.

The mechanism that describes the pigmentation of *Hibiscus* flower cultivars is poorly studied [4]. To explore the relationships of the color of the flowers, their antioxidant capacity and the content of flavonoids present in the petals, the Pearson correlation coefficients of the color parameters, the ORAC value and the content of anthocyanins, flavonols, flavan-3-ols, flavanonols, flavones, and flavanones were determined. These relationships are shown in a heatmap (Figure 5). The color was associated in the following way: the parameters a*, h and C were positively correlated with the ORAC value, the L∗ values were negatively correlated with the antioxidant capacity, and the parameter b* presented a small negative correlation. The clearer the shade of the flowers (white, yellow, or lilac), the higher the L* values and the lower the antioxidant capacity values of their extracts. The opposite effect was observed for flowers with darker shades (orange < pink < red), with lower L* values, higher antioxidant capacity, and high content of anthocyanins, flavan-3-ols and flavanonols in their extracts.

Studies have explored the relationship between the color and chemical composition of different flowers, e.g., roses [44], water lilies [49], daffodils [50], crabapples [51] and canola [52]. The results obtained in this comparative study of *H. rosa-sinensis* cultivars were consistent with those reported by Wan et al. [44], who studied the relationship between color and chemical composition of various rose cultivars and reported that the red cultivar had a higher concentration of anthocyanins compared to that of the yellow, orange, and pink cultivars. However, in the latter, the concentrations of carotenoids and flavonols were higher. The authors [44] reported that kaempferol and quercetin aglycones were the predominant flavonols in roses. Sheyth and De [53] reported that the red cultivar had the highest total phenolic content and antioxidant activity compared with the pink, yellow and white cultivars. According to Tai et al. [51], the highest concentration of anthocyanins in wild apple flowers was related to the amount of the enzyme chalcone synthase, which regulates anthocyanin biosynthesis and, therefore, was responsible for the red coloration and color variation in the petals of apple cultivars.

According to Yin et al. [52], the main components detected in the extracts of red and pink petals of *Brassica napus* were derivatives of epicatechin, quercetin and isorhamnetin, whereas those from yellow and white petals were found to be kaempferol derivatives. In the *Hibiscus* flower extracts studied here, glycosylated derivatives of quercetin were the major components in pink, orange, and yellow flowers. Yin et al. [52] reported that derivatives of hydroxycinnamic acid, sinapoyl malate, naringenin-7-*O*-glucoside, cyanidin-3-glucoside, cyanidin-3,5-di-*O*-glucoside, petunidin-3-*O*-β-glucopyranoside, isorhamnetin -3-*O*-glucoside, kaempferol-3-*O*-glucoside-7-*O*-glucoside, quercetin-3,4-*O*-di-glucopyranoside, quercetin-3-*O*-glucoside, and delphinidin-3-*O*-glucoside were responsible for the variety of colors in the petals of *Brassica napus*.

Flavan-3-ols (catechin and epicatechin) were mainly detected in the extracts of red cultivars. This can be explained through the biosynthetic pathway (Figure 6): the precursor of catechin is leucoanthocyanidin, from which the anthocyanidins, precursors of epicatechin, and anthocyanins are biosynthesized. The increased bioavailability of anthocyanidins can lead to the biosynthesis of flavan-3-ols. In the extracts from white flowers (Table 2), neither anthocyanins nor anthocyanidins were detected, and the concentration of flavan-3-ol was 120 times lower than that found in the extracts of red cultivars (Table 2).

Patel et al. [13] carried out a comparative study of the extracts of red, pink, orange, yellow and white cultivars of leaves, stems and roots of *Hibiscus rosa-sinensis* and found the highest and lowest total phenol content in the red and white cultivars, respectively. To the best of our knowledge, our research is the first comparative study of flavonoids present in extracts from different cultivars of *Hibiscus rosa-sinensis* flowers.

Anthocyanins, flavan-3-ol and flavanonols, among the flavonoids present in *H. rosa-sinensis* flowers, showed a high positive correlation with the ORAC value, and the higher content of these compounds, the higher the antioxidant capacity value measured by the ORAC method. Some glycosylated derivatives of quercetin and total quercetins also showed positive correlations with the ORAC value, but lower correlations compared with other substances. A negative correlation was observed for flavonols, anthocyanins, flavan-3-ols, and flavanonols: while the concentration of flavonols increased in the extracts obtained from pink, orange, yellow and white *Hibiscus* cultivars, these compounds decreased in the red cultivars.

The flavonol content, which was inversely proportional to the content of anthocyanins, flavan-3-ols and total flavanonols, was associated with lower antioxidant capacity. The extract of the red cultivar, with the highest content of anthocyanins, had a higher antioxidant capacity than the extract of the white flower, characterized by a high content of flavonols; however, its antioxidant capacity was the lowest compared with that of the other floral extracts studied.

## 4. Materials and Methods

### 4.1. Reagents

The following reagents and standard substances were used in this work: pelargonidine chloride (95%), cyanidin chloride (95%), naringenin (95%), kaempferol (90%), kaempferol-3-glucoside (95%), eriodictyol (95%), apigenin (95%), rutin (95%), quercetin (95%), quercetin-3-glucoside (98%), catechin (98%), epicatechin (97%), AAPH [2,2’-azinobis-(2-amidino-propane) dihydrochloride] (98%), α-tocopherol (97%), Trolox^®^ (6-hydroxy-2,5,7,8-tetramethyl-chroman-2-carboxylic acid) (97%), BHT (98.8%), and fluorescein [disodium 2-(3-oxo-6-oxido-3H-xanthene-9-yl benzoate)], purchased from Sigma-Aldrich (St. Louis, MO, USA). Cyanidin-3-glucoside (95%), cyanidine-3,5-glucoside (95%), cyanidin-3-rutinoside (99%), delphinidin-3-glucoside (90%), and pelargonidine-3-glucoside (95%) were purchased from PhytoLab GmbH & Co. KG (Vestenbergsgreuth, Germany). Methanol and absolute ethanol (99.9%), formic acid Suprapur (98–100%), potassium biphosphate (99%), hydrochloric acid (37%), acetonitrile or methanol grade LC/MS (≥99.9%) were obtained from Merck (Darmstadt, Germany).

### 4.2. Vegetal Material

Flowers from 16 cultivars of *Hibiscus rosa-sinensis* (Figure 1) were collected in the experimental garden of the CENIVAM Research Center on the main campus of Universidad Industrial de Santander, located in Bucaramanga, Santander, Colombia, (N 07 °08,422′ W 073°06,960′). The botanical identification of five cultivars of *H. rosa-sinensis* flowers was carried out at the Institute of Natural Sciences, Faculty of Sciences of the National University of Colombia (Bogotá campus) and was carried out by the botanist Dr. José Luis Fernández (Vouchers N° 579224–579228). The other flower cultivars were identified in the UIS Herbarium, School of Biology, Faculty of Sciences of the Industrial University of Santander (Vouchers N° 20881–20886).

### 4.3. Color Measurement

The color of the flowers of the *Hibiscus rosa-sinensis* cultivars (Figure 1) was measured on the adaxial surface of fresh petals with an STS-VIS microspectrophotometer (Ocean Optics, Dunedin, FL, USA.), size 40 × 42 × 24 mm, with wavelength ranges of λ = 190–650 nm (UV) and λ = 350–800 nm (Vis). The CIELAB (International Commission on Illumination) system was used. The colorimetric coordinates (L*, a*, b*, Cab* and hab) were obtained with Ocean View 1.5.0 software (Ocean Optics, Dunedin, FL, USA) After color measurement, the petals were extracted with solvent.

### 4.4. Solvent Extraction

Solvent extraction was performed according to Sierra et al. [57]. Three fresh petals of each *H. rosa-sinensis* cultivar (0.5 g) were immersed in an ethanol solution (20 mL, 0.5% HCl, 1:1 *v/v* in type I water) using an ultrasonic bath (15 min, 50 °C, Elma-sonic S15H, Singen, Germany). Each extract was filtered (Whatman N° 1 filter paper), and the residue was extracted once more. The *H. rosa-sinensis* extracts were rotoevaporated and then dried in a VirTis AdVantage Plus tray lyophilizer (Gardiner, NY, USA.) and stored at 7 °C in the absence of light prior to analysis.

### 4.5. UHPLC-ESI^+^-Orbitrap-MS Analysis

Extract analysis was performed on a liquid chromatograph coupled to a mass spectrometer with a heated electrospray ionization interface (HESI-II). Acquisition was performed in positive ion mode with an Orbitrap high-resolution mass detector (Exactive Plus, Thermo-Fisher Scientific, Sunnyvale, CA, USA). Chromatographic separation was carried out with a Hypersil GOLD aQ column (Thermo Scientific, Sunnyvale, CA, USA) of 100 mm, L, × 2.1 mm, id, × 1.9 μm particle size. The mobile phase used was A—formic acid/water and B—formic acid/acetonitrile, with a flow of 300 µL/min. Formic acid was used at 0.2% *v/v* in the mobile phase. The injection volume was 1 µL. The Vspray was 3.5 kV. The nebulizer temperature was set at 350 °C, and the capillary temperature was set at 320 °C. Nitrogen (>99%) was obtained in an NM32LA generator (Peak Scientific, Inchinnan, STK, UK). Spectra were initially acquired in full scan mode, the Orbitrap-HRMS resolution was set to 70,000 full width at half maximum, (*m/z* 200 RFWHM); automatic gain control (AGC) was 3 × 10^6^; and the maximum injection time to the C-trap was 200 ms. The protonated molecules were fragmented in the HCD cell using energies of 10, 20, 30, or 40 eV. Mass spectra were registered in AIF (All-Ion-Fragmentation) mode for each collision energy at an RFWHM of 35000, AGC of 3 × 10^6^ and an injection time of 50 ms in the C-Trap cell; all mass spectra were obtained in the range of *m/z* 80–1000. Calibration of the LC/MS instrument was performed using Pierce LTQ Velos ESI Positive Ion Solution (Thermo Scientific, Rockford, IL, USA).

Data were processed with Thermo XcaliburTM Roadmap software, Version 3.1.66.10. Compound identification was carried out by measuring the exact masses of [M]^+^ or [M+H]^+^ ions, finding the elemental composition of protonated molecules and their products, and comparing the retention times, t_R_, and mass spectra of each reference compound with those observed in the extracted ion chromatograms of the extracts. The metabolomic (METLINTM, http://metlin.scripps.edu/ (accessed on 1 June 2020 to 10 January 2023)) and phytochemistry (PCIDB, http://www.genome.jp./db/pcodb (accessed on 1 June 2020 to 10 January 2023)) databases were used. The quantification of the compounds present in the extracts was carried out by the external calibration method, and the results were expressed in cyanidin-3-glucoside equivalents for anthocyanins and in quercetin-3-glucoside equivalents and kaempferol-3-glucoside equivalents for the rest of the substances, except for catechin and epicatechin, whose standard compounds were used for their quantification.

### 4.6. Antioxidant Assay: Oxygen Radical Absorption Capacity

The oxygen radical absorbance capacity (ORAC) was determined for each extract following the methodologies described by Ou et al. [58] and Huang et al. [59], with some modifications. The antioxidant capacity was measured from the difference between the area under the curve (AUC) of each sample and the AUC of the blank. Fluorescein (FL) (150 µL, 8.16 × 10^−5^ mM) was added to the sample dilutions in phosphate buffer (pH 7.4), and the mixtures were incubated (37 °C, 18 min) followed by AAPH [2,2’-azinobis-(2-amidino-propane) dihydrochloride] addition (25 µL, 153 mM). The decrease in fluorescein fluorescence (FL) was recorded for 80 min using wavelengths of λ = 490 nm for excitation and λ = 510 nm for emission. Trolox^®^ (97%, Sigma-Aldrich, St. Louis, MO, USA) was used as a control standard. The net area obtained was the basis for calculating the μmol Trolox^®^/g of extract using a calibration curve. Determinations were made in triplicate on a Turner Biosystems Inc., ModulusTM II Microplate Multimode Reader (Sunnyvale, CA, USA with 96-well poly(styrene) microplates and a fluorescence module. They were expressed as the mean value ± the standard deviation (*n* = 3).

## 5. Conclusions

A positive correlation was found between the antioxidant capacity of the *H. rosa-sinensis* flower extracts and the content of anthocyanins, flavan-3-ols and total flavanonols, whereas the correlation was negative with flavonols. The antioxidant capacity and the highest amounts of anthocyanins, flavan-3-ols and flavononols were found in the extracts of the darker red flowers. *H. rosa-sinensis* red flowers can be a source of natural ingredients and bioactive compounds such as anthocyanins, flavan-3-ols and flavanonols for pharmaceutical and cosmetic products. Yellow and white *H. rosa-sinensis* cultivars can serve as natural sources for the extraction of flavonols. Knowing the composition of flavonoids present in flowers based on their color can serve as a basis for understanding in more detail the mechanisms that regulate color and its origin in flowers, particularly in cultivars of *Hibiscus* spp.

## Figures and Tables

**Figure 1 molecules-28-01779-f001:**
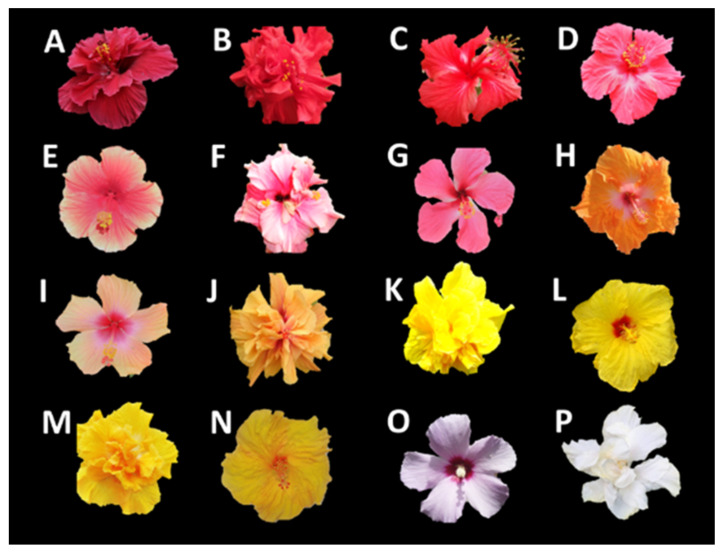
Flowers of 16 *H. rosa-sinensis* cultivars collected from the natural population of the CENIVAM experimental botanical garden, Universidad Industrial de Santander, Bucaramanga, Colombia. (**A**–**D**) Red. (**E**–**G**) Pink. (**H**–**J**) Orange. (**K**–**N**) Yellow. (**O**) Lilac. (**P**) White.

**Figure 2 molecules-28-01779-f002:**
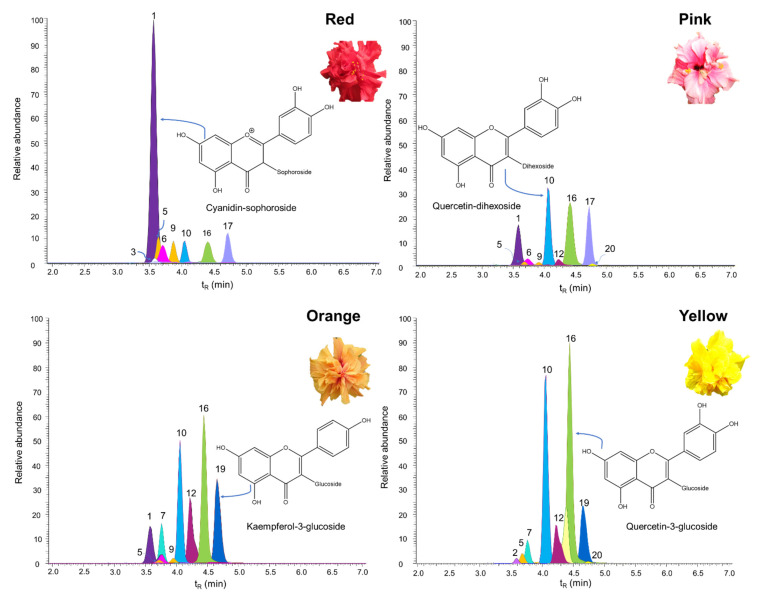
Chromatographic profiles obtained by LC/MS of the hydroalcoholic extracts of four *H*. *rosa*-*sinensis* cultivars. Chromatographic peak identification appears in Table 1.

**Figure 3 molecules-28-01779-f003:**
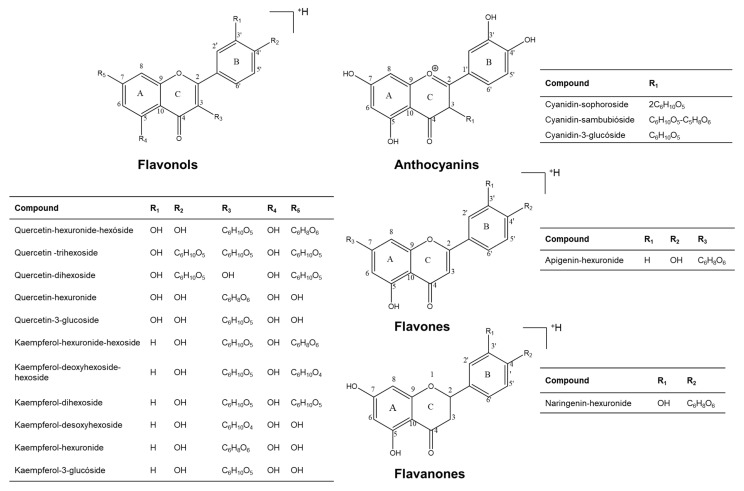
Chemical structures of flavonols, anthocyanins, flavones and flavanones detected in the *H*. *rosa*-*sinensis* flower extracts studied in this work.

**Figure 4 molecules-28-01779-f004:**
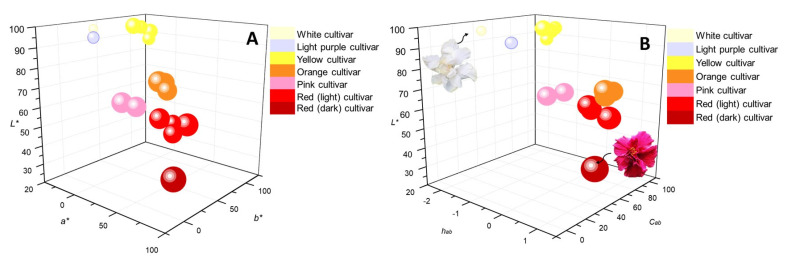
Three-dimensional bubble scatter graph represented with a color scale that corresponds to the floral extracts of *H. rosa-sinensis* studied, according to the CIELAB colorimetric parameters: (**A**). *L**, *a**, *b* and (**B**). *L**, *Cab*, *hab*. The bubble sizes are proportional to the value of the antioxidant activity, measured by the ORAC method.

**Figure 5 molecules-28-01779-f005:**
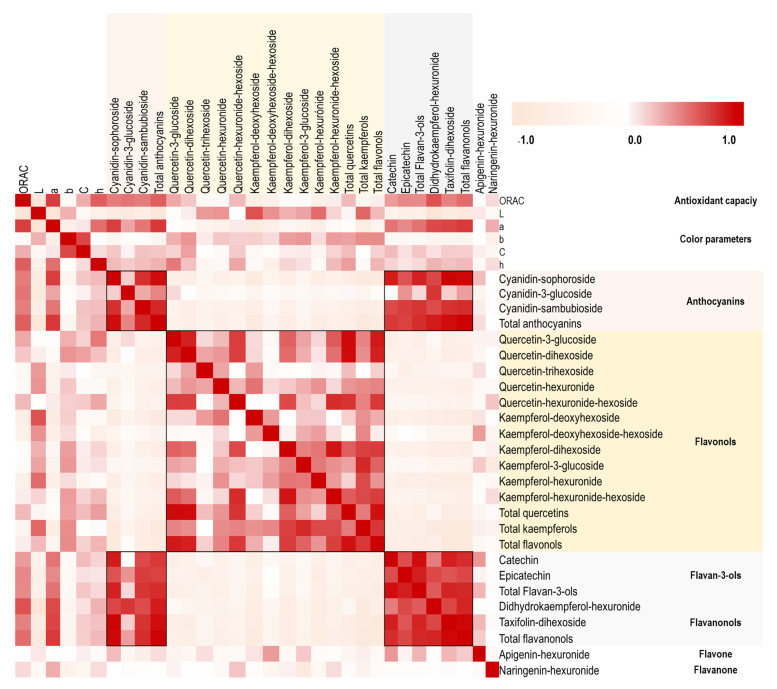
Heatmap of the correlation matrix of the color parameters, antioxidant capacity (ORAC) and 20 compounds present in the extracts of the 16 cultivars of *H. rosa-sinensis*. Each square indicates the Pearson correlation coefficient for a data pair, and the intensity of the red and beige colors in the heatmap indicates the level of positive and negative correlation, respectively.

**Figure 6 molecules-28-01779-f006:**
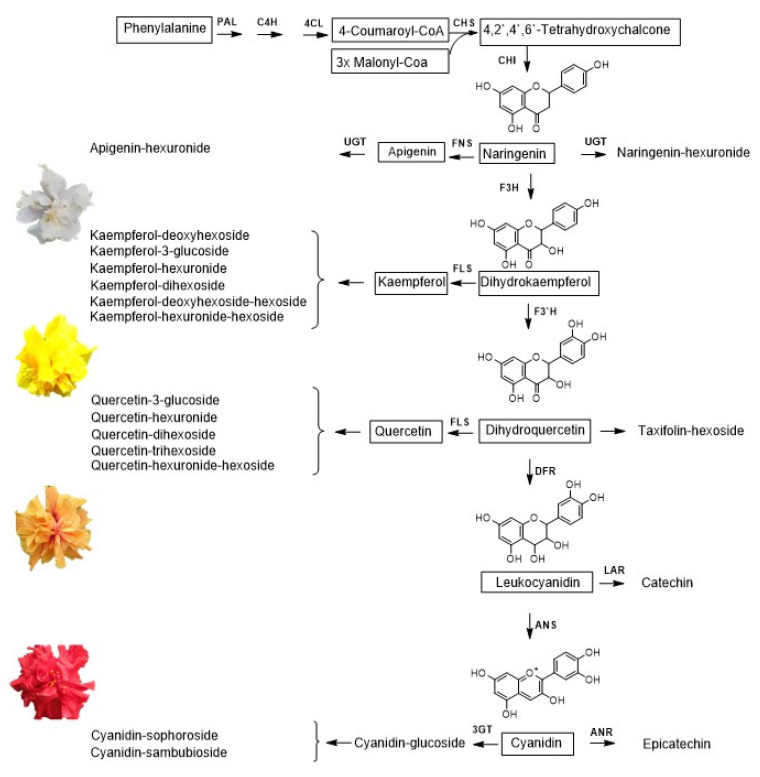
General scheme of flavonoid biosynthesis in *Hibiscus rosa-sinensis* flowers. PAL (phenylalanine-ammonium lyase), C4H (cinamate-4-hydroxylase), 4CL (4-coumarate CoA ligase), CHI (flavanone chalcone isomerase), FNS (Fla-vanone synthase), UGT (glucuronyltransferase), F3H (flavanone-3-hydroxylase), FLS (flavonol synthase), F3`H (flavonoid-3`hydroxylase), DFR (dihydroflavonol-4-reductase), LAR (leukucoanthocyanidin reductase), ANS (anothocyanin synthase), ANR (anthocyanidin reductase), 3GT (anthocyanidin-3-glucosyltransferase). Adapted from Hammerbacher et al. [54], Gerats, T. and Strommer [55] and Hong et al. [56].

**Table 1 molecules-28-01779-t001:** Exact masses obtained by UHPLC-ESI+-Orbitrap-MS of [M]^+^ ions or [M+H]^+^ protonated molecules of flavonoids found in *H. rosa-sinensis* flower extracts.

Fig 2 N°	Type	Compound	Formula	Calculated Mass		Exp. Mass	∆ ppm	HCD *, eV	Product Ions			Identification Criteria	Refs.
				[M]^+^	[M+H]^+^				Fragment	Formula	*m/z* (I,%)		
1	Anthocyanin	Cyanidin-sophoroside	C_27_H_31_O_16_	611.16066	-	611.16132	1.07	20	[M-2C_6_H_10_O_5_]^+^	C_15_H_11_O_6_	287.05493 (100%)	a,b	[22,25]
									[M-2C_6_H_10_O_5_-C_6_H_4_O_4_]^+^	C_9_H_7_O_2_	147.04404 (1%)		
									[M-2C_6_H_10_O_5_-C_8_H_6_O_3_]^+^	C_7_H_5_O_3_	137.02324 (1%)		
									[M-2C_6_H_10_O_5_-C_8_H_6_O_4_]^+^	C_7_H_5_O_2_	121.02866 (1%)		
2	Flavonol	Quercetin-hexuronide-hexoside	C_27_H_28_O_18_	-	641.13484	641.13501	0.26	20	[(M+H)-C_6_H_10_O_5_]^+^	C_21_H_19_O_13_	479.08209 (100%)	a,b	[26]
									[(M+H)-C_6_H_10_O_5_-C_6_H_8_O_6_]^+^	C_15_H_11_O_7_	303.04999 (43%)		
									[(M+H)-C_6_H_10_O_5_-C_6_H_8_O_6_-H_2_O]^+^	C_15_H_9_O_6_	285.03949 (2%)		
									[(M+H)-C_6_H_10_O_5_-C_6_H_8_O_6_-C_8_H_6_O_3_]^+^	C_7_H_5_O_4_	153.01836 (1%)		
3	Flavanonol	Taxifolin-dihexoside	C_27_H_32_O_17_	-	629.17122	629.17078	0.71	10	[(M+H)-C_6_H_10_O_5_]^+^	C_21_H_23_O_12_	467.11832 (15%)	a,b	[27]
									[(M+H)-2C_6_H_10_O_5_]^+^	C_15_H_13_O_7_	305.06561 (100%)		
									[(M+H)-2C_6_H_10_O_5_-H_2_O]^+^	C_15_H_11_O_6_	287.05496 (40%)		
									[(M+H)-2C_6_H_10_O_5_-C_8_H_8_O_3_]^+^	C_7_H_5_O_4_	153.01832 (2%)		
4	Anthocyanin	Cyanidin-sambubioside	C_26_H_29_O_15_	581.15009	-	581.15039	0.50	20	[M-C_6_H_10_O_5_-C_5_H_8_O_4_]^+^	C_15_H_11_O_6_	287.05508 (100%)	a,b	[25,28]
									[M-C_6_H_10_O_5_-C_5_H_8_O_4_-C_6_H_4_O_4_]^+^	C_9_H_7_O_2_	147.04404 (1%)		
									[M-C_6_H_10_O_5_-C_5_H_8_O_4_-C_8_H_6_O_3_]^+^	C_7_H_5_O_3_	137.02339 (66%)		
									[M-C_6_H_10_O_5_-C_5_H_8_O_4_-C_8_H_6_O_4_]^+^	C_7_H_5_O_2_	121.02853 (1%)		
5	Flavan-3-ol	Catechin	C_15_H_14_O_6_	-	291.08631	291.08636	0.17	10	[(M+H)-C_6_H_6_O_3_]^+^	C_9_H_9_O_3_	165.05470 (25%)	a,b,c	[18]
									[(M+H)-C_8_H_8_O_3_]^+^	C_7_H_7_O_3_	139.03903 (100%)		
									[(M+H)-C_8_H_8_O_4_]^+^	C_7_H_7_O_2_	123.04424 (47%)		
6	Anthocyanin	Cyanidin-3-glucoside	C_21_H_21_O_11_	449.10784	-	449.10818	0.77	20	[M-C_6_H_10_O_5_]^+^	C_15_H_11_O_6_	287.05515 (100%)	a,b,c	[20,25]
									[M-C_6_H_10_O_5_-C_8_H_6_O_3_]^+^	C_7_H_5_O_3_	137.02344 (25%)		
									[M-C_6_H_10_O_5_-C_8_H_6_O_4_]^+^	C_7_H_5_O_2_	121.02854 (1%)		
7	Flavonol	Kaempferol-hexuronide-hexoside	C_27_H_28_O_17_	-	625.13992	625.14044	0.82	20	[(M+H)-C_6_H_10_O_5_]^+^	C_21_H_19_O_12_	463.08743 (100%)	a,b	[26]
									[(M+H)-C_6_H_10_O_5_-C_6_H_8_O_6_]^+^	C_15_H_11_O_6_	287.05490 (45%)		
									[(M+H)-C_6_H_10_O_5_-C_6_H_8_O_6_-C_8_H_6_O_2_]^+^	C_7_H_5_O_4_	153.01833(1%)		
8	Flavonol	Quercetin-trihexoside	C_33_H_40_O_22_	-	789.20840	789.20880	0.50	20	[(M+H)-C_6_H_10_O_5_]^+^	C_27_H_31_O_17_	627.15594 (3%)	a,b	[29]
									[(M+H)-2C_6_H_10_O_5_]^+^	C_21_H_21_O_12_	465.10339 (60%)		
									[(M+H)-3C_6_H_10_O_5_]^+^	C_15_H_11_O_7_	303.05011 (100%)		
									[(M+H)-3C_6_H_10_O_5_-C_8_H_6_O_3_]^+^	C_7_H_5_O_4_	153.01836 (15%)		
9	Flavan-3-ol	Epicatechin	C_15_H_14_O_6_	-	291.08631	291.08626	0.14	10	[(M+H)-C_6_H_6_O_3_]^+^	C_9_H_9_O_3_	165.05473 (25%)	a,b,c	[18]
									[(M+H)-C_8_H_8_O_3_]^+^	C_7_H_7_O_3_	139.03902 (100%)		
									[(M+H)-C_8_H_8_O_4_]^+^	C_7_H_7_O_2_	123.04424 (47%)		
10	Flavonol	Quercetin-dihexoside	C_27_H_30_O_17_	-	627.15557	627.15613	0.88	10	[(M+H)-C_6_H_10_O_5_]^+^	C_21_H_21_O_12_	465.10281 (16%)	a,b	[19,29]
									[(M+H)-2C_6_H_10_O_5_]^+^	C_15_H_11_O_7_	303.04990 (100%)		
									[(M+H)-2C_6_H_10_O_5_-C_8_H_6_O_3_]^+^	C_7_H_5_O_4_	153.01819 (1%)		
11	Flavonol	Kaempferol-deoxyhexoside-hexoside	C_27_H_30_O_15_	-	595.16574	595.16632	0.96	10	[(M+H)-C_6_H_10_O_5_]^+^	C_21_H_21_O_10_	433.11310 (20%)	a,b	[21]
									[(M+H)-C_6_H_10_O_5_-C_6_H_10_O_4_]^+^	C_15_H_11_O_6_	287.05502 (100%)		
									[(M+H)-C_6_H_10_O_5_-C_6_H_10_O_4_-C_8_H_6_O_2_]^+^	C_7_H_5_O_4_	153.01822(1%)		
12	Flavonol	Kaempferol-dihexoside	C_27_H_30_O_16_	-	611.16066	611.16132	1.01	10	[(M+H)-C_6_H_10_O_5_]^+^	C_21_H_21_O_11_	449.10754 (20%)	a,b	[21]
									[(M+H)-2C_6_H_10_O_5_]^+^	C_15_H_11_O_6_	287.05508 (100%)		
									[(M+H)-2C_6_H_10_O_5_-H_2_O]^+^	C_15_H_9_O_5_	269.04465 (1%)		
									[(M+H)-2C_6_H_10_O_5_-C_8_H_6_O_2_]^+^	C_7_H_5_O_4_	153.01814 (1%)		
13	Flavonol	Kaempferol-deoxyhexoside	C_21_H_20_O_10_	-	433.11292	433.11292	0.68	20	[(M+H)-C_6_H_10_O_4_]^+^	C_15_H_11_O_6_	287.05508 (100%)	a,b	[21]
									[(M+H)-C_6_H_10_O_4_-C_8_H_6_O_2_]^+^	C_7_H_5_O_4_	153.01820 (1%)		
14	Flavonol	Quercetin-hexuronide	C_21_H_18_O_13_	-	479.08201	479.08203	0.03	20	[(M+H)-C_6_H_8_O_6_]^+^	C_15_H_11_O_7_	303.04968 (100%)	a,b	[30]
									[(M+H)-C_6_H_8_O_6_-H_2_O]^+^	C_15_H_9_O_6_	285.03899 (1%)		
									[(M+H)-C_6_H_8_O_6_-C_8_H_6_O_3_]^+^	C_7_H_5_O_4_	153.01825 (1%)		
15	Flavanonol	Didhydrokaempferol-hexuronide	C_21_H_20_O_12_	-	465.10275	465.10297	0.45	20	[(M+H)-C_6_H_8_O_6_]^+^	C_15_H_13_O_6_	289.07037 (100%)	a,b	[29]
									[(M+H)-C_6_H_8_O_6_-C_8_H_8_O_2_]^+^	C_7_H_5_O_4_	153.01805 (10%)		
16	Flavonol	Quercetin-3-glucoside	C_21_H_20_O_12_	-	465.10275	465.10300	0.52	10	[(M+H)-C_6_H_10_O_5_]^+^	C_15_H_11_O_7_	303.05005 (100%)	a,b,c	[18]
									[(M+H)-C_6_H_10_O_5_-H_2_O]^+^	C_15_H_9_O_6_	285.03960 (1%)		
									[(M+H)-C_6_H_10_O_5_-C_8_H_6_O_3_]^+^	C_7_H_5_O_4_	153.01799 (1%)		
17	Flavanone	Naringenin-hexuronide	C_21_H_20_O_11_	-	449.10784	449.10770	0.31	10	[(M+H)-C_6_H_8_O_6_]^+^	C_15_H_13_O_5_	273.07559 (100%)	a,b	[31]
									[(M+H)-C_6_H_8_O_6_-C_8_H_8_O]^+^	C_7_H_5_O_4_	153.01829(8%)		
18	Flavonol	Kaempferol-hexuronide	C_21_H_18_O_12_	-	463.08670	463.08710	0.86	20	[(M+H)-C_6_H_8_O_6_]^+^	C_15_H_11_O_6_	287.05490 (45%)	a,b	[21]
									[(M+H)-C_6_H_8_O_6_-C_8_H_6_O_2_]^+^	C_7_H_5_O_4_	153.01833(1%)		
19	Flavonol	Kaempferol-3-glucoside	C_21_H_20_O_11_	-	449.10784	449.10784	0.56	10	[(M+H)-C_6_H_10_O_5_]^+^	C_15_H_11_O_6_	287.04718 (100%)	a,b,c	[18,32]
									[(M+H)-C_6_H_10_O_5_-C_8_H_6_O_2_]^+^	C_7_H_5_O_4_	153.01855 (4%)		
20	Flavone	Apigenin-hexuronide	C_21_H_18_O_11_	-	447.09218	447.09216	0.05	20	[(M+H)-H_2_O]^+^	C_21_H_17_O_10_	429.08162 (1%)	a,b	[31]
									[(M+H)-C_6_H_8_O_6_]^+^	C_15_H_11_O_5_	271.05997(45%)		
									[(M+H)-C_6_H_8_O_6_-C_8_H_6_O]^+^	C_7_H_5_O_4_	153.01824 (17%)		
									[(M+H)-C_6_H_8_O_6_-C_8_H_6_O_3_]^+^	C_7_H_5_O_2_	121.02857 (1%)		

a Tentative identification based on [M]^+^ ions or protonated molecules [M+H]^+^ reported in the literature for the genus *Hibiscus* spp. [18,19,20,21]. b Tentative identification based on the study of exact masses of ions and fragments, fragmentation patterns, scientific article data and spectral (METLIN^TM^, http://metlin.scripps.edu/) and phytochemical (PCIDB, http://www.genome.jp./db/pcodb accessed on from 1 June 2020 to 10 January 2023) databases. c Confirmatory identification based on retention indices, mass spectra (ESI^+^-HRMS), and their comparison with those of standard compounds. * HCD: Higher-energy collision dissociation cell.

**Table 2 molecules-28-01779-t002:** Total flavonoids, organized by subfamilies, and CIELAB colorimetric parameters measured for different *H. rosa-sinensis* cultivars studied.

Cultivar	CIELAB Parameters (Value ± S, *n* = 5)	Total Flavonoids (mg Flavonoid/g Extract) (Value ± S, *n* = 3)	Antioxidant Activity	Color *
Sample	Code Figure 1	*L**	*a**	*b**	*C*_ab_*	*h_ab_*	Anthocyanins	Flavonols	Flavan-3-ols	Flavanonols	Flavones	μmol Trolox^®^/gExtract, ±S, *n* = 3
Red	A	28 ± 2	65 ± 4	35 ± 4	73 ± 2	0.5 ± 0.07	19.9 ± 0.9	14.9 ± 0.9	0.91 ± 0.01	1.48 ± 0.05	0.07 ± 0.001	3840 ± 74	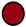
Red	B	54 ± 1	65 ± 4	53 ± 2	84 ± 3	0.7 ± 0.04	23.6 ± 0.6	5.7 ± 0.3	2.5 ± 0.1	2.0 ± 0.2	0.31 ± 0.02	3480 ± 64	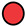
Red	C	61 ± 1	64 ± 2	24 ± 2	68 ± 2	0.4 ± 0.01	21.8 ± 0.9	8.0 ± 0.4	2.3 ± 0.1	2.0 ± 0.1	0.53 ± 0.04	3200 ± 100	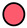
Red	D	56 ± 3	77 ± 3	15 ± 0.5	78 ± 3	0.2 ± 0.01	19.8 ± 0.9	6.2 ± 0.5	1.3 ± 0.4	1.9 ± 0.1	0.5 ± 0.2	2900 ± 133	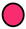
Pink	E	58 ± 2	69 ± 0.3	26 ± 1	74 ± 0.6	0.4 ± 0.02	6.6 ± 0.1	15.8 ± 0.7	0.23 ± 0.01	0.59 ± 0.03	0.18 ± 0.03	2900 ± 126	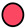
Pink	F	69 ± 3	41 ± 3	−6 ± 0.8	42 ± 3	−0.2 ± 0.03	3.4 ± 0.2	19 ± 1	0.31 ± 0.05	0.32 ± 0.01	0.41 ± 0.01	3200 ± 132	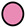
Pink	G	69 ± 2	57 ± 0.4	−7 ± 2	57 ± 0.2	−0.1 ± 0.03	9.8 ± 0.5	17.8 ± 0.6	1.9 ± 0.1	0.95 ± 0.05	0.17 ± 0.04	3130 ± 33	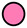
Orange	H	73 ± 1	35 ± 3	51 ± 2	62 ± 2	1.0 ± 0.03	1.6 ± 0.1	34 ± 1	0.14 ± 0.01	0.11 ± 0.01	0.53 ± 0.03	3250 ± 40	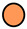
Orange	I	69 ± 3	44 ± 3	51 ± 3	68 ± 3	0.9 ± 0.04	1.7 ± 0.1	28 ± 1	0.23 ± 0.04	0.12 ± 0.02	0.16 ± 0.01	3200 ± 135	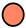
Orange	J	72 ± 2	37 ± 3	58 ± 1	68 ± 2	1.0 ± 0.03	3.0 ± 0.1	32 ± 1	0.34 ± 0.03	0.09 ± 0.001	0.13 ± 0.01	3000 ± 77	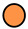
Yellow	K	93 ± 1	−8.1 ± 0.1	93 ± 0.3	93 ± 0.3	−1.5 ± 0.01	-	29 ± 1	0.22 ± 0.02	-	0.17 ± 0.03	2580 ± 53	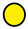
Yellow	L	96 ± 2	−16 ± 0.8	77 ± 2	79 ± 2	−1.4 ± 0.01	-	23.9 ± 0.5	0.050 ± 0.002	-	0.25 ± 0.01	2440 ± 90	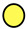
Yellow	M	95 ± 3	−18 ± 2	91 ± 0.8	93 ± 1	−1.4 ± 0.01	-	19.2 ± 0.8	0.92 ± 0.02	-	0.56 ± 0.02	2100 ± 99	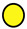
Yellow	N	90 ± 1	−7 ± 0.5	89 ± 2	89 ± 2	−1.5 ± 0.01	0.30 ± 0.01	16.0 ± 0.5	0.20 ± 0.03	-	0.22 ± 0.01	1930 ± 72	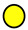
Light purple	O	96 ± 2		5 ± 1	6 ± 1	−0.1 ± 0.03	1.1 ± 0.1	12.3 ± 0.6	0.35 ± 0.04	0.18 ± 0.01	0.21 ± 0.01	1850 ± 87	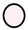
White	P	99 ± 5	−4 ± 1	9 ± 2	10 ± 2	−1.1 ± 0.09	-	12.6 ± 0.6	0.021 ± 0.003	-	0.28 ± 0.02	1580 ± 41	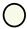

* Chromatic representation of RGB coordinates, obtained with the application http://colorizer.org. (accessed on from 20 January 2021 to 15 December 2022).

**Table 3 molecules-28-01779-t003:** Phenolic compound content (mg/g) in the hydroalcoholic *H. rosa-sinensis* flower extracts of cultivars studied in this work.

Figure 2 N°	Compound	Cultivar, mg/g of Extract, (Value ± S, *n* = 3)											
		Red			Orange			Pink			Yellow		
1	Cyanidin-sophoroside ^1^	21	±	1	2.0	±	0.1	2.4	±	0.2		-	
2	Quercetin-hexuronide-hexoside ^2^	0.1	±	0.01	1.0	±	0.1	0.40	±	0.04	0.50	±	0.06
3	Taxifolin-dihexoside ^2^	1.5	±	0.2		-		0.20	±	0.01		-	
4	Cyanidin-sambubioside^1^	1.1	±	0.1	-				-			-	
5	Catechin	1.5	±	0.1	0.20	±	0.02	0.20	±	0.04	0.20	±	0.02
6	Cyanidin-3-glucoside	1.5	±	0.2	1.1	±	0.1	1.0	±	0.1		-	
7	Kaempferol-hexuronide-hexoside ^3^		-		1.70	±	0.02	0.20	±	0.02	0.60	±	0.06
8	Quercetin-trihexoside		-			-		0.20	±	0.01	0.10	±	0.01
9	Epicatechin	1.1	±	0.1	0.20	±	0.01	0.20	±	0.01		-	
10	Quercetin-dihexoside ^2^	1.5	±	0.2	8.2	±	0.9	6.8	±	0.2	8.20	±	0.08
11	Kaempferol-deoxyhexoside-hexoside ^3^	0.20	±	0.01	0.20	±	0.01	0.50	±	0.01	0.30	±	0.04
12	Kaempferol-dihexoside ^3^		-		3.1	±	0.3	0.60	±	0.01	1.30	±	0.01
13	Kaempferol-deoxyhexoside ^3^	0.30	±	0.01	0.50	±	0.01	1.50	±	0.01	2.60	±	0.06
14	Quercetin-hexuronide ^2^	0.20	±	0.01	0.70	±	0.01	0.70	±	0.03	1.20	±	0.09
15	Didhydrokaempferol-hexuronide ^3^	0.10		0.01		-		0.30		0.02		-	
16	Quercetin-3-glucoside	2.0	±	0.03	11	±	1	7	±	1	11.0	±	0.5
17	Naringenin-hexuronide ^2^	2.40	±	0.15		-		7.0	±	0.6		-	
18	Kaempferol-hexuronide ^3^		-		0.80	±	0.04	0.20	±	0.01	0.9	±	0.1
19	Kaempferol-3-glucoside		-		5.1	±	0.3		-		2.7	±	0.3
20	Apigenin-hexuronide ^2^	0.30	±	0.02	0.10	±	0.01	0.40	±	0.01	0.20	±	0.01

^1^ Expressed as cyanidin-3-glucoside equivalents. ^2^ Expressed as quercetin-3-glucoside equivalents. ^3^ Expressed as kaempferol-3-glucoside equivalents.

**Table 4 molecules-28-01779-t004:** Antioxidant activity of plant extracts with anthocyanins, flavonols, flavan-3-ols, flavanonols and flavones.

Species	Family	Components	Assay	Result			
				Values *			Units
*Punica granatum* [38]	Lythraceae	Anthocyanins (delphinidin, cyanidin, and pelargonidin)	SOD	17	±	1	SOD-equivalent units/mg of freezedried extract
*Solanum tuberosum* [35]	Solanaceae	Anthocyanins, flavonoids and phenolic acids	ORAC	28.25	-	250.67	µmol Trolox^®^/g of extract
*Ginkgo biloba* [39]	Lamiaceae	Flavonols (quercetin-3-rutinoside, quercetin-3-*O*-rhamnosyl-rhamnosyl-glucoside, and kaempferol-3-*O*-rhamnosyl-rhamnosyl-glucoside)	ORAC	13.18	±	0.24	µmol Trolox^®^/g of extract
*Pelargonium graveolens* [40]	Geraniaceae	Flavonols and tannins	ABTS^+•^	131.54	-	241.83	μg Trolox equivalents/mg of extract
*Vitis vinifera* [41]	Vitaceae	Flavan-3-ols (catechin and epicatechin), flavonol (quercetin), phenolic acids (gallic and ellagic acids), stilbenoid (resveratrol) and ascorbic acid	ORAC	450.51	±	74.02	µmol Trolox^®^/g of extract
*Lippia graveolens* [42]	Verbenaceae	Flavanonol (taxifolin), flavones (apigenin-7-O-glucoside and hispidulin), dihydrochalcone (phlorizin), flavanones (eriodictyol, naringenin and pinocembrin), *O*-methylated flavone (genkwanin) and flavonols (quercetin and galangin)	ORAC	1.66	±	0.284	mg Trolox^®^/mL of extract
*Artemisia annua* [43]	Asteraceae	Flavone (apigenin), flavonol (rutin) and phenolic acid (caffeic acid)	ORAC	736.26	±	17.55	µM Trolox^®^/g of plant

* Values taken literally from the references [35,38,39,40,41,42,43]. SOD: superoxide radical scavenging activity. ABTS^+•^: radical scavenging assay.

## Data Availability

The supporting data are included in the data repository of the CIBIMOL research group from Universidad Industrial de Santander (Bucaramanga, Colombia).

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
