# Peer review of "Color, Antioxidant Capacity and Flavonoid Composition in Hibiscus rosa-sinensis Cultivars"

_molecules, 2023, doi:10.3390/molecules28041779_

Round 1

Reviewer 1 Report

In this manuscript the authors present the potential relationship between the type and quantity of flavonoids and the color and antioxidant capacity of extracts from H. rosa-sinensis flowers from 16 cultivars. The paper is interesting and there is a lot of experimental and analytical work, but it lacks clarity in some sections:

There is no discussion about the results represented in figure 2. For example, which are the compounds that are repeated in the 4 cultivars? Is the color linked to the compounds that are more abundant?

It is necessary to correct the format of table 2, the colored circles are offset from their corresponding samples

Specify how performance was calculated. Is it calculated based on the total solids that can be extracted? Or based on the initial weight of the sample and the solid content in the extract obtained?

Why were fresh petals used for extraction instead of dried petals? The extraction process is more efficient if it is carried out with dry samples.

I recommend adding some references

Rengarajan, S., Melanathuru, V., Govindasamy, C., Chinnadurai, V., & Elsadek, M. F. (2020). Antioxidant activity of flavonoid compounds isolated from the petals of Hibiscus rosa sinensis. Journal of King Saud University-Science, 32(3), 2236-2242.

Sheth, F., & De, S. (2012). Evaluation of comparative antioxidant potential of four cultivars of Hibiscus rosa-sinensis L. by HPLC-DPPH method. Free Radicals and Antioxidants, 2(4), 73-78.

Vankar, P. S., & Srivastava, J. (2008). Comparative study of total phenol, flavonoid contents and antioxidant activity in Canna indica and Hibiscus rosa sinensis: Prospective natural food dyes. International journal of food engineering, 4(3).

Sri Raghavi, R., Visalakshi, M., Karthikeyan, S., Amutha Selvi, G., Thamaraiselvi, S. P., & Gurusamy, K. (2022). Standardisation of anthocyanin extraction techniques from hibiscus (Hibiscus rosa-sinensis) petals for biocolour utilisation.

Author Response

  1. There is no discussion about the results represented in figure 2. For example, which are the compounds that are repeated in the 4 cultivars? Is the color linked to the compounds that are more abundant?

Compounds detected in all H. rosa-sinensis hydroalcoholic extracts were included in the manuscript (Section 2.1). The contribution of each compound to the color of four H. rosa-sinensis cultivars was presented in Figure 5 (Discussion section). In red flowers the color is related to anthocyanins, which were not detected in yellow and white flowers whose color is mainly related to flavonols.

  1. It is necessary to correct the format of table 2, the colored circles are offset from their corresponding samples

The changes in Table 2 were applied.

  1. Specify how performance was calculated. Is it calculated based on the total solids that can be extracted? Or based on the initial weight of the sample and the solid content in the extract obtained?

The information was included in the discussion section.

“The solvent extraction yield was calculated by dividing the hydroalcoholic extract weight over the plant material weight”.

  1. Why were fresh petals used for extraction instead of dried petals? The extraction process is more efficient if it is carried out with dry samples.

The plant material for extraction was used fresh because a color-chemical composition relationship was being investigated. The color was measured first and the extraction was carried out immediately.

  1. I recommend adding some references

References were included in the manuscript.

  1. Sri Raghavi, R.; Visalakshi, M.; Karthikeyan, S.; Amutha Selvi, G.; Thamaraiselvi, S.; Gurusamy, K. Standardisation of anthocyanin extraction techniques from hibiscus (Hibiscus rosa-sinensis) petals for biocolour utilisation. J. Pharm. Innov. 2022, 11, 303-309.
  2. Vankar, P.; Srivastava, J. Comparative study of total phenol, flavonoid contents and antioxidant activity in Canna indica and Hibiscus rosa sinensis: prospective natural food dyes. Int. J. Food Eng. 2008, 4, 1-15.
  3. Rengarajan, S.; Melanathuru, V.; Govindasamy, C.; Chinnadurai, V.; Elsadek, M. Antioxidant activity of flavonoid compounds isolated from the petals of Hibiscus rosa sinensis. J. King Saud Univ. Sci. 2020, 32, 2236-2242.
  4. Sheth, F.; De, S. Evaluation of comparative antioxidant potential of four cultivars of Hibiscus rosa-sinensis L. by HPLC-DPPH method. Free Radic. Antioxid. 2012, 2, 73-78.

Reviewer 2 Report

Dear Authors

ِYou have succeeded to determine the phenolic compounds of the Hibiscus rosa-sinensis in different cultivars completely. An interesting work of this researcher is that you have found the relationship between color and these 4 groups of phenolic compounds (anthocyanin, flavanol, flavanone and flavonoid), as well as the relationship between antioxidant properties and these 4 groups of polyphenols. In my opinion, in order to complete the conclusion of this very interesting research, I suggest to prepare a table of the relationship between antioxidant property and these four phenolic groups in other plants based on the results of similar literatures to conclude whether in other plant genus also this relationship exists whether no, this relationship based on the plant genus can be different.

Author Response

Indeed, the relationship between antioxidant property and chemical structure has been an active field of research, carried out at different levels. Flavonoids and phenolic compounds have been the subject of many of these investigations. The plant genus does not play a significant role in this relationship. It is strongly dependent on chemical structure aspects such as the accessibility of the phenolic group and steric effects due to neighbor substituents. In the manuscript, the table and references with the antioxidant activity of extracts from other plant species were added.

  1. Noda, Y.; Kaneyuki, T.; Mori, A.; Packer, L. Antioxidant activities of pomegranate fruit extract and its anthocyanidins: delphinidin, cyanidin, and pelargonidin. J. Agric. Food Chem. 2002, 50, 166-171.
  2. Andre, C.; Ghislain, M.; Bertin, P.; Oufir, M.; Del Rosario Herrera, M.; Hoffmann, L.; Evers, D. Andean potato cultivars (Solanum tuberosum L.) as a source of antioxidant and mineral micronutrients. J. Agric. Food Chem. 2007, 55, 366-378.
  3. Zheng, W.; Wang, S. Antioxidant activity and phenolic compounds in selected herbs. J. Agric. Food Chem. 2001, 49, 5165-5170.
  4. Ben Elhadj, I.; Tajini, F.; Boulila, A.; Jebri, M.; Boussaid, M.; Messaoud, C.; Sebaï, H. Bioactive compounds from tunisian Pelargonium graveolens (L’Hér) essential oils and extracts: α-amylase and acethylcholinesterase inhibitory and antioxidant, antibacterial and phytotoxic activities. Ind. Crops Prod. 2020, 158, 1-11.
  5. Yilmaz, Y.; Toledo, R. Major flavonoids in grape seeds and skins: antioxidant capacity of catechin, epicatechin, and gallic acid. J. Agric. Food Chem. 2004, 52, 255-260.
  6. Cortés-Chitala, M.; Flores-Martínez, H.; Orozco-Ávila, I.; León-Campos, C.; Suárez-Jacobo, Á.; Estarrón-Espinosa, M.; López-Muraira, I. Identification and quantification of phenolic compounds from Mexican oregano (Lippia graveolens HBK) hydroethanolic extracts and evaluation of its antioxidant capacity. Molecules. 2021, 26, 2-18.
  7. Skowyra, M.; Gallego, M.; Segovia, F.; Almajano, M. Antioxidant properties of Artemisia annua extracts in model food emulsions. Antioxidants. 2014, 3, 116-128.

Reviewer 3 Report

The color, antioxidant capacity and flavonoid of Hibiscus rosa-sinensis cultivars were investigated and compared by measuring their colorimetric parameters and identifying the chemical composition of extracts obtained from petals. The research is interesting and the writing is good. The results could help to understand the mechanisms that regulate color and its origin in flowers, and make use of this plant. Minor revision is needed, questions and suggestions are as following.

1. Different kinds of flavonoids, including anthocyanins, were identified and quantified in the extracts. Do they contain other phenolic compositions? For example, phenolic acids may also contribute to antioxidant activity.

2.  Generally, more than 3 assays should be carried out to study antioxidant capacity of flower extract samples. I suggest the authors to refer to some related papers, DOI: 10.3390/foods8100471, and discuss in this study.

3. The extracts were obtained by acidified ethanol solution, which is generally regarded as the optimal extraction solvents for anthocyanins. While, for flavonoids, acetone is better, how to choose extraction solvent in this study?

4. Are there any carotenoids in the petal?

5. Line 300-301, is obscure, please make a rearrangement for this sentence.

Author Response

  1. Different kinds of flavonoids, including anthocyanins, were identified and quantified in the extracts. Do they contain other phenolic compositions? For example, phenolic acids may also contribute to antioxidant activity.

Phenolic acids were not identified by UHPLC-ESI+ -Orbitrap-MS LC/MS in H. rosa-sinensis flowers extracts. Phenolic acids have antioxidant activity but their contributions as pigments in H. rosa-sinensis flowers are still unknown.

  1. Generally, more than 3 assays should be carried out to study antioxidant capacity of flower extract samples. I suggest the authors to refer to some related papers, DOI: 10.3390/foods8100471, and discuss in this study.

The selection of the ORAC method as the only method to determine the antioxidant activity of the samples was added in the manuscript with the suggested reference [35].

  1. Xiang, J.; Yang, C.; Beta, T.; Liu, S.; Yang, R. Phenolic profile and antioxidant activity of the edible tree peony flower and underlying mechanisms of preventive effect on H2O2-induced oxidative damage in Caco-2 cells. Foods. 2019, 8, 1-17.
  2. Prior, R.; Wu, X.; Schaich, K. Standardized methods for the determination of antioxidant capacity and phenolics in foods and dietary supplements. J. Agric. Food Chem. 2005, 53, 4290-4302.
  3. The extracts were obtained by acidified ethanol solution, which is generally regarded as the optimal extraction solvents for anthocyanins. While, for flavonoids, acetone is better, how to choose extraction solvent in this study?

Ethanol solution was used as a solvent extraction because other conventional solvents (methanol and acetone) are restricted to obtain natural ingredients for cosmetic and pharmaceutical applications.

  1. Are there any carotenoids in the petal?

Carotenoids were not determined in H. rosa-sinensis flowers extracts under study, but carotenoids contribution as a flower pigments has been reported by Wan et al. [41].

  1. Wan, H.; Yu, C.; Han, Y.; Guo, X.; Luo, L.; Pan, H.; Zheng, T.; Wang, J.; Cheng, T.; Zhang, Q. Determination of flavonoids and carotenoids and their contributions to various colors of rose cultivars (Rosa spp.). Front. Plant Sci. 2019, 10, 1-14.
  2. Line 300-301, is obscure, please make a rearrangement for this sentence.

The text in lines 300 to 301 was modified.